# The Prognostic Role of Different Blood Cell Count-to-Lymphocyte Ratios in Patients with Lung Cancer at Diagnosis

**DOI:** 10.3390/cancers17233879

**Published:** 2025-12-04

**Authors:** Ourania Papaioannou, Oraianthi Fiste, Eva Theohari, Fotios Sampsonas, Foteinos-Ioannis Dimitrakopoulos, Angelos Koutras, Ioannis Gkiozos, Ioannis Vathiotis, Elias Kotteas, Argyrios Tzouvelekis

**Affiliations:** 1Department of Respiratory Medicine, University Hospital of Patras, University of Patras, 26504 Patras, Greece; ouraniapapaioannou@outlook.com (O.P.); theohari.eva@gmail.com (E.T.); argyris.tzouvelekis@gmail.com (A.T.); 2Oncology Unit, 3rd Department of Medicine, “Sotiria” Hospital for Diseases of the Chest, National and Kapodistrian University of Athens, 11527 Athens, Greece; ofiste@med.uoa.gr (O.F.); yiannisgk@hotmail.com (I.G.); johnvathiotis1@gmail.com (I.V.); ilkotteas@med.uoa.gr (E.K.); 3Department of Oncology, University Hospital of Patras, University of Patras, 26504 Patras, Greece; fodimitrak@yahoo.gr (F.-I.D.); angkoutr@upatras.gr (A.K.); 4Department of Pulmonary, Critical Care and Sleep Medicine, Yale School of Medicine, New Haven, CT 06511, USA

**Keywords:** lung cancer, prognostic biomarkers, NLR, PLR, MLR

## Abstract

Our study investigates potential associations of general blood cell count parameters, including neutrophil to lymphocyte ratio (NLR), platelet to lymphocyte ratio (PLR), and monocyte to lymphocyte ratio (MLR) at baseline (diagnosis) with the risk of mortality in patients with lung cancer (LC). We clearly demonstrated that high NLR and PLR cut-off values at diagnosis could reliably differentiate survivors from non-survivors in a cohort of patients with LC. On the other hand, MLR levels failed to serve as predictors of mortality. Our analysis revealed that NLR, PLR, or MLR values were independent of TNM staging at diagnosis and smoking history. Our study provides solid evidence that parameters of peripheral blood could complement already established markers of disease severity and progression to reliably predict clinical outcomes in pre-treated patients with LC.

## 1. Introduction

Lung cancer (LC) represents the leading cause of cancer-related mortality for both males and females, imposing a substantial medical and economic burden not only for patients but also for their relatives. The estimated numbers of new invasive cancer cases and deaths in the United States of America in 2023 were 238,340 for males and 127,070 for females [1]. Based on histopathology, non-small-cell LC (NSCLC) and small-cell LC (SCLC) are the main histological subcategories. NSCLC is more frequent and is further classified into adenocarcinoma (adNSCLC), squamous cell carcinoma (sqNSCLC), and large-cell carcinoma (LCLC) [2]. Recently, the treatment management for advanced and metastatic LC has dramatically changed. Targeted therapy in patients with a molecular driver mutation or antibody-directed immunotherapy against specific target molecules, such as programmed death-1 (PD-1), its ligand (PD-L1), and the cytotoxic T-lymphocyte-associated protein 4 receptor (CTLA-4), in combination or not with traditional chemotherapy, has remarkably ameliorated survival rates at present [3]. Despite the aforementioned therapeutic advances, the prognosis of LC patients is still poor. Personalized treatment decisions within a multidisciplinary framework are of utmost importance. Yet, it is more than evident that there is difficulty in predicting outcomes for each individual patient, in terms of precision oncology, mainly due to disease complexity [4,5]. Therefore, there is still an amenable need for clinically applicable and validated biomarkers in patients with LC.

Inflammation has been suggested as a prognostic biomarker in cancer due to its contribution as a hallmark in the tumor microenvironment. Inflammatory immune cells are increasingly accepted to be major components of tumors. These inflammatory cells operate in conflicting directions, either antagonizing or promoting neoplastic lesions [6]. As a consequence, systemic inflammation is present in various phases of carcinogenesis. These stages include tumor initiation, promotion, progression, invasion, and metastasis, with the inflammation playing a crucial role as a key prognostic determinant in cancer [7,8,9]. Despite the fact that several studies have discussed the prognostic contribution of the acute phase inflammatory markers, such as lactate dehydrogenase (LDH), erythrocyte sedimentation rate, alanine transaminase (ALT), aspartate transaminase (AST), interleukins, and C-reactive protein (CRP), none of these parameters are specific to carcinogenesis [10]. On the other side, systemic inflammation can be estimated on a regular basis through simple, inexpensive, easily accessible, and clinician-friendly inflammatory determinants based on the complete blood count (CBC) parameters, such as neutrophil to lymphocyte ratio (NLR), platelet to lymphocyte ratio (PLR), and monocyte to lymphocyte ratio (MLR). These blood cell count-to-lymphocyte ratios are considered to be independent prognostic factors in several types of malignancy, particularly breast and lung cancer [11,12,13,14,15].

The prognostic role of complete blood count parameters in patients with various chronic lung diseases has been thoroughly investigated, and previous studies have found quite promising, yet still exploratory, results [16,17,18]. With regard to LC, biomarkers of systemic inflammatory response, such as plasma CRP, and general blood cell count parameters, including absolute white blood cells or their components, seem to add prognostic granularity to the clinical, histopathological, and radiological parameters of tumor development and progression [19,20]; yet data needs to be further validated, and prognostic cut-off thresholds need to be better defined [21,22,23]. In light of this observation, real-life data supporting the prognostic role of general blood cell count parameters in patients with LC are sorely needed. In this vein, we conducted a retrospective study to investigate the potential associations of general blood cell count parameters, including NLR, PLR, and MLR, at baseline (diagnosis) with the risk of mortality in patients with LC.

## 2. Methods and Materials

### 2.1. Study Design and Patient Selection

In this retrospective study, between 1 June 2020 and 31 May 2024, we recorded consecutive patients who presented to the Department of Respiratory Medicine, University Hospital of Patras, Patras, Greece, and received a first diagnosis of LC. Diagnosis was based on histopathological examination of tissue biopsies, acquired through endobronchial ultrasound transbronchial needle biopsy (EBUS-TBNB) or computed tomography (CT)-guided fine needle biopsy (FNB). Patients with CBC and TNM staging at diagnosis were included in the analysis, while patients with active infection confirmed through clinical, laboratory, or radiological examination at the time of diagnosis were excluded, as well as patients with uncontrolled cardiovascular or pulmonary disease, endocrinological disorders, and a history of other types of cancer. Data collection and analysis were approved by the Institutional Review Board and the Local Ethics Committee (protocol number 8055/17-03-2025). Informed consent was obtained from all individual participants included in the study. Age, smoking history, histopathology results, molecular driver mutations, PD-L1 expression, ΤΝΜ staging, CBC at diagnosis, and vital status in May 2025 were recorded. All included patients were treated based on multidisciplinary discussions between oncologists, pulmonologists, pathologists, radiation oncologists, and thoracic surgeons, taking into account the current treatment guidelines at the time of diagnosis.

### 2.2. Outcome Measures

The primary outcome was mortality risk based on NLR, PLR, or MLR at diagnosis. Secondary outcomes included comparison of ΤΝΜ staging and smoking with NLR, PLR, and MLR at diagnosis.

### 2.3. Statistical Analysis

With regard to baseline data, summary descriptive statistics were generated with categorical data displayed as absolute numbers and relative frequencies. Following the Kolmogorov–Smirnov test for normality, continuous data were presented as mean ± standard deviation (SD) or medians with a 95% Confidence Interval (95% CI). A Mann–Whitney U or *t*-test was used for the investigation of differences between groups based on the absence or presence of normality. Kaplan–Meier curves were used to present the primary outcome, and cumulative incidence curves were compared between the two groups. In particular, we examined the prognostic accuracy of median levels of studied parameters (NLR, PLR, MLR) in differentiating high- from low-risk groups of patients with LC in terms of all-cause mortality. More specifically, patients were split based on the median value of each parameter into high- and low-risk groups. Mortality risk was compared between high- and low-risk groups. In addition, the multivariate Cox regression Hazard ratio models were used to investigate the associations between the studied parameters and mortality. The covariates considered here included age, gender, smoking, TNM staging, histopathology, molecular driver mutations, and PD-L1 expression. *p*-values < 0.05 were considered statistically significant. The data were analyzed using MedCalc v19.4.1(MedCalc Software Ltd., Ostend, Belgium).

## 3. Results

### 3.1. Baseline Characteristics

We identified 353 patients with a diagnosis of LC. Baseline characteristics are summarized in Table 1. The mean age ± SD at the time of diagnosis was 68.1 ± 9.1 years. Most patients were male (77.9%, n = 275) and current or ex-smokers (58.1%, n =205 and 39.1%, n =138, respectively). Histological diagnosis was non-small-cell lung cancer (NSCLC), small-cell lung cancer (SCLC), and not otherwise specified (NOS) in 67.1% (n = 237), 29.8% (n = 105), and 3.1% (n = 11) of patients, respectively. Adenocarcinoma NSCLC was more common (40.2%, n = 142) compared to squamous NSCLC (25.5%,n = 90). In 12.9% of patients, we identified EGFR, KRAS, ALK, or BRAF molecular driver mutations, while PD-L1 expression was positive in 20.7% of patients. The majority of enrolled patients presented with advanced stage IV LC at diagnosis (63.2%, n = 223).

### 3.2. Mortality Risk and Secondary Outcomes

Kaplan–Meier curves showed that patients with higher than the median NLR and PLR values at diagnosis presented with significantly higher mortality risk compared to patients with lower than the median [HR: 0.58, (95% CI: 0.42 to 0.81) *p* = 0.0009, HR: 0.71, (95% CI: 0.53 to 0.95) *p* = 0.02, respectively], while no associations with the risk of mortality and higher versus lower than the median MLR were observed [HR: 0.84, (95% CI: 0.63 to 1.12) *p* = 0.22] (Figure 1A–C). The complementary subgroup NLR, PLR, MLR median values based on histology and stage are presented in Table 2. With regard to secondary outcomes, univariate analysis revealed no differences in terms of mortality between higher vs. lower than the median NLR, PLR, and MLR groups and: (1) TNM staging at diagnosis [4.0 (95% CI: 4.0–4.0) vs. 4.0 (95% CI: 4.0–4.0), *p* = 0.95, 4.0 (95% CI: 4.0–4.0) vs. 4.0 (95% CI: 4.0–4.0), *p* = 0.09, 4.0 (95% CI: 4.0–4.0) vs. 4.0 (95% CI: 4.0–4.0), *p* = 0.4, respectively] (Figure 2A–C); (2) smoking pack-years [70 (95% CI: 60–80) vs. 80 (95% CI: 60–80), *p* = 0.10, 70 (95% CI: 60–80) vs. 80 (95% CI: 60–80), *p* = 0.46, 80 (95% CI: 60–80) vs. 70 (95% CI: 60–80), *p* = 0.96, respectively] (Figure 3A–C). Following adjustment for age, gender, smoking, TNM staging, histopathology, molecular driver mutations, and PD-L1 expression, the multivariate model revealed that increased NLR and advanced TNM staging were independent risk factors for mortality (Figure 4).

## 4. Discussion

Inflammation plays a crucial role in tumor development and progression across several cancer types, including LC [6,8,24]. Inflammatory cells, including neutrophils, lymphocytes, monocytes, and platelets, not only represent major sources of several pro-inflammatory cytokines, chemokines, and growth factors, thus contributing to tumor progression and metastasis, but also reflect the body’s systemic inflammatory response to cancer, and, therefore, may serve as prognosticators of disease progression and survival [22]. Nevertheless, studies linking CBC parameters to clinical outcomes in LC patients are relatively scarce and require further validation. In line with this notion, we conducted a real-life retrospective study evaluating the ability of peripheral blood parameters to reflect pre-treatment clinical outcomes in a cohort of newly diagnosed patients with LC.

Our study exhibits some major attributes, as the analysis clearly demonstrated that high-NLR and -PLR cut-off values at diagnosis could reliably differentiate survivors from non-survivors in a cohort of patients with LC. On the other hand, MLR levels failed to serve as predictors of mortality. Our analysis revealed that NLR, PLR, and MLR values were independent of TNM staging at diagnosis and smoking history. Our findings are consistent with those of previous reports, indicating a possible link between parameters of CBC and prognosis in patients with LC.

In line with this notion, research of Wu et al. revealed that tumor-cell migration, invasion, and metastasis are promoted by M2 tumor-associated macrophages, which produce growth factors and enhance angiogenesis, tissue remodeling, and repair [25,26,27,28]. NLR is a pivotal regulator of systemic inflammation and a potential prognostic biomarker in LC as well [29]. There is a broad consensus that elevated levels of NLR are indicative of an aberrant lymphocyte-mediated immune response to malignancy [30]. On the other hand, an increased number of circulating neutrophils has been associated with the upregulation of vascular endothelial growth factor, the master regulator of angiogenesis, leading to tumor growth and metastasis [30]. Importantly, studies have shown that patients with LC, colon cancer, gastric cancer, and other solid malignancies and higher NLR or PLR values generally have a worse prognosis [31]. The predictive role of PLR seems to be contradictory among patients with NSCLC and SCLC, which could be attributed to different cohorts, sample sizes, stages of disease, and therapeutic regimens [32,33,34,35]. In addition, neutrophils, monocytes, and lymphocytes are commonly associated with prognosis [36]. It is well known that monocytes promote angiogenesis, immune tolerance, and tumor spread, and simultaneously exert strong phagocytic activity [37]. On the other hand, a decrease in lymphocyte levels leads to immunological suppression through relevant mediators [38]. As a consequence, MLR could represent a rather promising biomarker of prognosis in LC [21,39,40,41,42,43,44,45].

Emerging data show that high NLR, high PLR, and high MLR at baseline (prior to any therapeutic intervention) are significantly associated with poor overall survival (OS) [21]. More specifically, in a single-center Australian study including 279 patients with stage IV NSCLC actively treated from 2010 to 2015, the analysis suggested that increased values of NLR and PLR were associated with an increased hazard of death, while increased values of LMR were associated with a reduced hazard of death [21]. In line with this notion, Bar-Ad et al. showed that patients with elevated pre-treatment NLR exhibited a statistically significant worse OS [22]. NLR and PLR have not only served as prognostic but also as theragnostic biomarkers, considering an association of elevated NLR and PLR values with worse OS and progression-free survival (PFS), and worse response rates in patients with metastatic NSCLC receiving nivolumab, excluding other confounding factors [46]. These results were further corroborated by Liu et al., who demonstrated that patients with advanced NSCLC and lower NLR and PLR exhibited a favorable prognosis [39].

The lack of reliable and clinically readable biomarkers of disease progression, survival, and treatment response is even more pronounced in SCLC cases. So far, besides TNM staging and patient performance status, patients’ genetic background, occupational history, and inflammation status have also been related to poor prognosis for SCLC cases [47,48]. Inflammatory indices have been applied to predict clinical outcomes in patients with SCLC, yet the results of several studies are characterized by inconsistency [49,50,51,52]. Grieshober et al. showed that among SCLC cases, those with pre-diagnosis NLR in the highest quartile exhibited increased mortality compared to those in the lowest quartile [53]. The aforementioned study provided preliminary data indicative of the potential use of pre-diagnosis CBCs in heavy smokers at high risk of developing lung cancer as useful patient-level information with regard to risk stratification [53]. To note that with regard to the association between smoking and NLR and PLR, the medical literature has shown that NLR increases in correlation with pack-years, while PLR is not affected or decreased in smokers [54,55]. Based on a meta-analysis by Zhou et al., PLR showed promising predictive performance in SCLC patients [35]. Indeed, patients with high PLR had shorter OS compared with those with lower levels, in both univariate and multivariate analyses [35]. The prognostic significance of LMR in SCLC has also been investigated [56]. In the whole patient cohort of a previous randomized study, PFS and OS were significantly shorter in the low LMR group. Further multivariate analysis showed that low LMR at diagnosis was an independent negative prognostic factor [56]. The clinical significance of LMR in patients treated surgically for SCLC was studied recently [57]. Higher LMR values before surgical intervention, analyzed through a multivariate Cox proportional hazard model, were considered a positive prognostic factor in terms of improved OS and disease-free survival (DFS) in patients with SCLC elective for surgery [57].

The interest concerning clinically applicable biomarkers in chronic lung diseases recently transformed our research interest with respect to inflammation indexes and their prognostic value in patients with LC. An accumulating body of evidence shows that inflammation and cancer are linked, as cellular mediators of inflammation are important key points of the local tumor environment [8,58,59,60]. The hallmarks of cancer-related inflammation include the presence of inflammatory cells and inflammatory mediators in tumor tissues, tissue remodeling, and angiogenesis similar to that seen in chronic inflammatory responses, and tissue repair [58]. The constitution of the tumor microenvironment—beyond cancer cells and their surrounding stroma—includes several innate and adaptive immune cells [59]. As a consequence, inflammation can be assessed systemically by measuring various hematological or biochemical markers in routine blood tests or as ratios of the aforementioned measurements, aiming to associate systemic inflammation markers with cancer risk at diagnosis [61,62]. More specifically, the following ratios, including NLR, PLR, and MLR based on peripheral lymphocyte, neutrophil, monocyte, and platelet counts, have been previously studied by previous studies in terms of morbidity and mortality [63,64]. Being a lethal malignancy of significant prevalence worldwide, LC presents an active area of preclinical and clinical studies investigating the prognostic potential of the easily accessible NLR, PLR, and MLR.

Despite relative enthusiasm arising from the above observations, our study is characterized by limitations. Firstly, our study presents inherent disadvantages of a retrospective real-life study. This registry was not designed to elucidate mechanistic pathways. Nevertheless, the role of real-world studies is to contribute additional information on prognostic biomarkers beyond highly selective patient cohorts in the context of randomized controlled trials. In addition, our sample size was moderate compared to previous, already-published studies for the prognostic role of CBC rates; yet the size is acceptable for a real-world study. Moreover, despite the fact that our research lacks novelty, as the prognostic role of CBC parameters in LC has been previously investigated in several study designs, the findings in the treatment of naïve patients remain relevant and could be useful in everyday clinical practice.

## 5. Conclusions

Our study provides solid evidence that parameters of peripheral blood could complement already established markers of disease severity and progression to reliably predict clinical outcomes in pre-treated patients with LC. Our data support the notion that NLR and PLR could be implemented in the everyday clinical management of patients with LC. Further prospective multicenter studies are sorely needed to identify optimal cut-off thresholds that could be extensively applied on both prognostic and theragnostic levels.

## Figures and Tables

**Figure 1 cancers-17-03879-f001:**
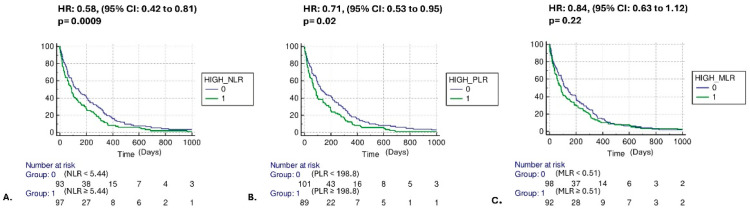
Kaplan–Meier curves for mortality risk. Patients with higher than the median NLR at diagnosis presented with significantly higher mortality risk compared to patients with lower than the median NLR [HR: 0.58, (95% CI: 0.42 to 0.81) *p* = 0.0009] (**A**); Patients with higher than the median PLR at diagnosis presented with significantly higher mortality risk compared to patients with lower than the median PLR [HR: 0.71, (95% CI: 0.53 to 0.95) *p* = 0.02] (**B**); Patients with higher than the median MLR at diagnosis were not presented with significantly higher mortality risk compared to patients with lower than the median MLR [HR: 0.84, (95% CI: 0.63 to 1.12) *p* = 0.22] (**C**). It bears noting that the vertical gap between the aforementioned curves decreases as time goes by. A small vertical gap means a small difference in the proportion of subjects who have survived up to that time.

**Figure 2 cancers-17-03879-f002:**
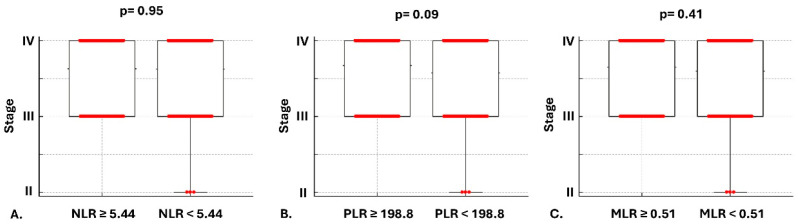
Comparison of ΤΝΜ staging with NLR, PLR, and MLR at diagnosis. Stage was not significantly more advanced in patients with higher than the median NLR at diagnosis compared to patients with lower than the median NLR [4.0 (95% CI: 4.0–4.0) vs. 4.0 (95% CI: 4.0–4.0), *p* = 0.95] (**A**); Stage was not significantly more advanced in patients with higher than the median PLR at diagnosis compared to patients lower than the median PLR [4.0 (95% CI: 4.0–4.0) vs. 4.0 (95% CI: 4.0–4.0), *p* = 0.09] (**B**); Stage was not significantly more advanced in patients with higher than the median MLR at diagnosis compared to patients with lower than the median MLR [4.0 (95% CI: 4.0–4.0) vs. 4.0 (95% CI: 4.0–4.0), *p* = 0.41] (**C**).

**Figure 3 cancers-17-03879-f003:**
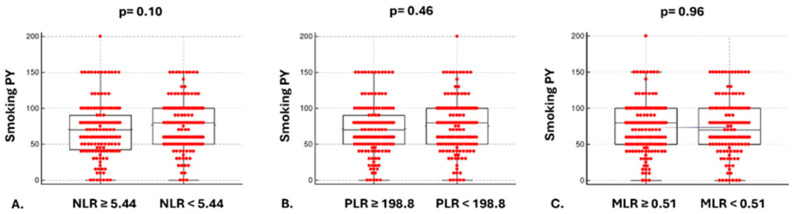
Comparison of smoking PY with NLR, PLR, and MLR at diagnosis. Smoking pack-years were not significantly higher in patients with higher than the median NLR at diagnosis compared to patients with lower than the median NLR [70 (95% CI: 60–80) vs. 80 (95% CI: 60–80), *p* = 0.10] (**A**); Smoking pack-years were not significantly higher in patients with higher than the median PLR at diagnosis compared to patients with lower than the median PLR [70 (95% CI: 60–80) vs. 80 (95% CI: 60–80), *p* = 0.46] (**B**); Smoking pack-years were not significantly higher in patients with higher than the median MLR compared to patients with lower than the median MLR [80 (95% CI: 60–80) vs. 70 (95% CI: 60–80), *p* = 0.96] (**C**).

**Figure 4 cancers-17-03879-f004:**
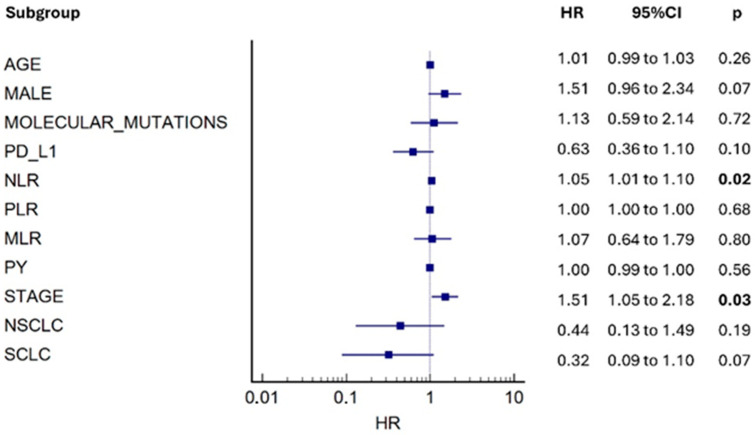
Multivariate Cox regression survival analysis for patients diagnosed with LC. Following adjustment for age, gender, smoking, TNM staging, histopathology, molecular driver mutations, and PD-L1 expression, the multivariate model revealed that increased NLR and advanced TNM staging were independent risk factors for mortality.

**Table 1 cancers-17-03879-t001:** Baseline characteristics.

Characteristics	(N, %)
Total number of patients	353
Age ± SD	68.1 ± 9.1
Male/Female	275 (77.9%)/78 (22.1%)
Current smokers/Ex-smokers/	205 (58.1%)/138 (39.1%)/
Never smokers	10 (2.8%)
NSCLC/SCLC/NOS	237 (67.1%)/105 (29.8%)/11 (3.1%)
adNSCLC/sqNSCLC	142 (40.2%)/90 (25.5%)
EGFR/KRAS/ALK/BRAF mutations	9 (2.5%)/31 (8.8%)/3 (0.8%)/3 (0.8%)
PD-L1 expression TPS: <1%/1–49%/≥50%	47 (13.3%)/32 (9.1%)/41 (11.6%)
Stage IIA/IIB/IIIA/IIIB/IIIC/IVA/IVB	1 (0.3%)/2 (0.6%)/31 (8.7%)/94 (26.6%)/2 (0.6%)/80 (22.7%)/143 (40.5%)

**Abbreviations**: adNSCLC: adenocarcinoma non-small-cell lung cancer; ALK: anaplastic lymphoma kinase, BRAF: B-Raf proto-oncogene serine/threonine kinase, EGFR: epidermal growth factor receptor, KRAS: Kirsten rat sarcoma virus, NOS: not otherwise specified, NSCLC: non-small-cell lung cancer, PD-L1: programmed cell death ligand 1, SCLC: small-cell lung cancer, SD: standard deviation, sqNSCLC: squamous non-small-cell lung cancer, TPS: tumor proportion score.

**Table 2 cancers-17-03879-t002:** Subgroup NLR, PLR, and MLR median values.

	NLR (%95CI)	PLR (95%CI)	MLR (95%CI)
NSCLC	5.84 (5.16 to 6.48)	205.5 (189.4 to 244.1)	0.52 (0.48 to 0.57)
SCLC	5.05 (4.40 to 5.56)	182.3 (157.5 to 213.2)	0.49 (0.42 to 0.53)
adNSCLC	5.80 (4.85 to 6.54)	207.8 (176.6 to 244.8)	0.53 (0.47 to 0.58)
sqNSCLC	5.69 (4.96 to 6.66)	201.7 (184.4 to 252.2)	0.52 (0.48 to 0.61)
Stage IIIA/IIIB/IIIC	5.56 (4.88 to 6.21)	187.2 (174.9 to 235.5)	0.50 (0.46 to 0.57)
Stage IVA/IVB	5.30 (4.98 to 6.14)	205.5 (188.6 to 229.2)	0.51 (0.48 to 0.56)

**Abbreviations:** adNSCLC: adenocarcinoma non-small-cell lung cancer, CI: confidence interval, NSCLC: non-small-cell lung cancer, SCLC: small-cell lung cancer, sqNSCLC: squamous non-small-cell lung cancer.

## Data Availability

Data is available on request. OP and AT have full access to the data and are the guarantors for these data.

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
