# Peer review of "The Prognostic Role of Different Blood Cell Count-to-Lymphocyte Ratios in Patients with Lung Cancer at Diagnosis"

_cancers, 2025, doi:10.3390/cancers17233879_

Round 1

Reviewer 1 Report

Comments and Suggestions for Authors

The authors of this article report data on NLR, PLR, and MLR in 353 patients collected over a 4-year period and analyze their relationship to clinical status and predictive significance for survival in patients with LC.This article received a large amount of clinical data and a long follow-up. The analysis method was reasonable, the results were credible, and the conclusions were reliable. It is of great practical significance in predicting the survival of patients with LC. It is a high-quality study.

Author Response

Thank you for your valuable comments and the fast peer-review process. Comments really helped us to improve the quality of our manuscript. 

Our point-by-point reply to reviewers follows:

Reviewer: 1

The authors of this article report data on NLR, PLR, and MLR in 353 patients collected over a 4-year period and analyze their relationship to clinical status and predictive significance for survival in patients with LC. This article received a large amount of clinical data and a long follow-up. The analysis method was reasonable, the results were credible, and the conclusions were reliable. It is of great practical significance in predicting the survival of patients with LC. It is a high-quality study.

AU: Thank you for your comment. We really appreciate it.

Reviewer 2 Report

Comments and Suggestions for Authors

The authors conducted a retrospective study on 353 newly diagnosed lung cancer patients to evaluate whether blood-based inflammatory markers (NLR, PLR, and MLR) at diagnosis predict mortality. They found that higher NLR and PLR, but not MLR, were associated with an increased risk of death, suggesting their potential value as simple prognostic biomarkers. However, there are a few things that need to be addressed before consideration. Below are my comments:

1) Truncate the title as it looks too lengthy

2) The introduction needs to be comprehensively discussed, following lung cancer and prognostic biomarkers, in a detailed manner relevant to this study.

3) Also, add more literature survey and cite relevant references in the introduction section"

4) Section 2 must be changed to "Methods and Materials

5) Add consumables and other details - a table can be added to discuss the same

6) Merge the Results and Discussion sections into a single section

7) In Figure 3, add the legends and captions more clearly

8) Discuss the challenges and limitations 

9) Add more recent relevant references to support the propsoed wrok

10) Please check the English language throughout the manuscript for grammatical errors.

Author Response

Thank you for your valuable comments and the fast peer-review process. Comments really helped us to improve the quality of our manuscript. 

Our point-by-point reply to reviewers follows:

Reviewer: 2

The authors conducted a retrospective study on 353 newly diagnosed lung cancer patients to evaluate whether blood-based inflammatory markers (NLR, PLR, and MLR) at diagnosis predict mortality. They found that higher NLR and PLR, but not MLR, were associated with an increased risk of death, suggesting their potential value as simple prognostic biomarkers. However, there are a few things that need to be addressed before consideration. Below are my comments:

Point 1.

Truncate the title as it looks too lengthy.

AU: Thank you for your comment. We modified the title as follows:

“The prognostic role of different blood cell count to lymphocyte ratios in patients with lung cancer at diagnosis.”

Point 2.

The introduction needs to be comprehensively discussed, following lung cancer and prognostic biomarkers, in a detailed manner relevant to this study.

AU: Thank you for your comment. We added the following in introduction section:

“Inflammation has been suggested as prognostic biomarker in cancer due to its contribution as hallmark in tumor microenvironment. Immune  cells are increasingly accepted to be major components of tumor lesions. These inflammatory cells operate in conflicting directions, either antagonizing or promoting neoplastic lesions. As a consequence, systemic inflammation has been implicated in various stages of carcinogenesis, including the initiation, progression, invasion, and metastatic phases, highlighting its potential role as a key prognostic determinant in cancer. Despite the fact that several studies have discussed the prognostic importance of the acute phase reactants in systemic inflammation, including lactate dehydrogenase (LDH), erythrocyte sedimentation rate, alanine transaminase (ALT), aspartate transaminase (AST), interleukins, and C-reactive protein (CRP), none of these parameters are specific to carcinogenesis. On the other side, systemic inflammation can be estimated on a regular basis through simple, inexpensive, easily accessible, and clinicians’ friendly  inflammatory markers based on the complete blood count (CBC) parameters such as neutrophil to lymphocyte ratio (NLR), platelet to lymphocyte ratio (PLR), and monocyte to lymphocyte ratio (MLR). These ratios are recognized as independent prognostic factors in several types of cancer, particularly lung and breast cancer.”

Point 3.

Also, add more literature survey and cite relevant references in the introduction section".

AU: Thank you for your comment.

We added the following references in introduction section:

  1. Hanahan D, Weinberg RA. Hallmarks of cancer: the next generation. Cell. 2011;144(5):646-74.
  2. Akcam TI, Tekneci AK, Turhan K, Duman S, Cuhatutar S, Ozkan B, et al. Prognostic value of systemic inflammation markers in early stage non-small cell lung cancer. Sci Rep. 2025;15(1):33886.
  3. Grivennikov SI, Greten FR, Karin M. Immunity, inflammation, and cancer. Cell. 2010;140(6):883-99.
  4. Elinav E, Nowarski R, Thaiss CA, Hu B, Jin C, Flavell RA. Inflammation-induced cancer: crosstalk between tumours, immune cells and microorganisms. Nat Rev Cancer. 2013;13(11):759-71.
  5. Tekneci AK, Akcam TI, Kavurmaci O, Ergonul AG, Ozdil A, Turhan K, et al. Relationship between survival and erythrocyte sedimentation rate in patients operated for lung cancer. Turk Gogus Kalp Damar Cerrahisi Derg. 2022;30(3):381-8.
  6. Ethier JL, Desautels D, Templeton A, Shah PS, Amir E. Prognostic role of neutrophil-to-lymphocyte ratio in breast cancer: a systematic review and meta-analysis. Breast Cancer Res. 2017;19(1):2.
  7. Li A, Mu X, He K, Wang P, Wang D, Liu C, Yu J. Prognostic value of lymphocyte-to-monocyte ratio and systemic immune-inflammation index in non-small-cell lung cancer patients with brain metastases. Future Oncol. 2020;16(30):2433-44.
  8. Winther-Larsen A, Aggerholm-Pedersen N, Sandfeld-Paulsen B. Inflammation scores as prognostic biomarkers in small cell lung cancer: a systematic review and meta-analysis. Syst Rev. 2021;10(1):40.
  9. Liao S, Sun H, Lu H, Wu J, Wu J, Wu Z, et al. Neutrophil-to-lymphocyte ratio-based prognostic score can predict outcomes in patients with advanced non-small cell lung cancer treated with immunotherapy plus chemotherapy. BMC Cancer. 2025;25(1):697.
  10. Jin CX, Liu YS, Qin HN, Teng YB, Sun R, Ma ZJ, et al. Peripheral inflammatory factors as prognostic predictors for first-line PD-1/PD-L1 inhibitors in advanced non-small cell lung cancer. Sci Rep. 2025;15(1):11206.

Point 4.

Section 2 must be changed to "Methods and Materials.

AU: Thank you for your comment. We modified it as requested.

Point 5.

Add consumables and other details - a table can be added to discuss the same

AU: Thank you for your comment. We did not use specific consumables during this study, as it was a retrospective registry based on medical records.

Point 6.

Merge the Results and Discussion sections into a single section

AU: Thank you for your comment. Manuscript sections were adjusted based on journal’s template.

Point 7.

In Figure 3, add the legends and captions more clearly

AU: Thank you for your comment. We modified it accordingly.  

Point 8.

Discuss the challenges and limitations 

AU: Thank you for your comment. Challenges and limitations are discussed in more detail in relevant section.

“Despite relative enthusiasm arising from the above observations, our study has some limitations. First of all, our study presents with inherent weaknesses of a retrospective real-life study. This registry was not designed to provide mechanistic data. Nevertheless, real-world studies can provide complementary data on prognostic biomarkers beyond highly-selective randomized controlled trial patient populations. Secondly, our sample size is moderate compared to previous reports for the prognostic role of CBC rates; yet, the size is acceptable for a real-life study. Moreover, despite the fact that our research lacks novelty, as the prognostic role of CBC parameters in LC has been previously investigated in several study designs, the findings in treatment naïve patients remain contemporary and could be useful in everyday clinical practice.”

Point 9.

Add more recent relevant references to support the proposed work

AU: Thank you for your comment. We updated our reference list.

Point 10.

Please check the English language throughout the manuscript for grammatical errors.

AU: Thank you for your comment. We rechecked the manuscript for grammatical errors

Reviewer 3 Report

Comments and Suggestions for Authors

In this manuscript, the authors have performed a retrospective study on the association between lung cancer and increased NLR and PLR. the study was performed on 357 patients. The authors analyzed the blood parameters and the lung cancer stages. The idea is good but the authors need more evidence to support the association study. Considering that  the current study is a retrospective study, I would suggest the authors to discuss the implication of this correlation in real life scenario where the lung cancer isnot detected yet. In many other diseases also there is an increase in PLR. In this context, I would suggest the authors to discuss the relevance of the study. Although the findings arenot novel, the findings will be useful for the society. My comments are provided below

Minor

1. The authors need to elaborate on the figure legend. The Table 1 is fine but all the figures need elaboration.  

Major

  1. In figure 1, the difference between two groups (for NLR and PLR) is very small. Although the statistical significance is quite strong, I would suggest the authors to discuss the biological relevance of this small difference.
  2. In Figure 3, the authors need to provide the rationale for studying the association between smoking and NLR.
  3. The Figure 4 needs elaboration. What is the biological meaning of the highlighted p value in Fig 4?
  4. The authors have discussed the limitation of the study but the authors didnot discuss its relevance in many other diseases. Is this specific for lung cancer. What is the exclusion criteria? Is there any comorbidity? 

Author Response

Thank you for your valuable comments and the fast peer-review process. Comments really helped us to improve the quality of our manuscript. 

Our point-by-point reply to reviewers follows:

Reviewer: 3

Dear Editor and Authors,

Thank you for asking me to review this manuscript titled "The Prognostic Role of Neutrophil to Lymphocyte Ratio, Platelet to Lymphocyte Ratio, and Monocyte to Lymphocyte Ratio in Patients with Lung Cancer at Diagnosis" by Dr. Ourania Papaioannou and colleagues from various institutions in Greece.

The whole concept of NLR, PLR and MLR in relation to cancer outcome has been done at nauseum!! with multiple and larger studies having been reported. The outcomes of this study not surprisingly are quite common and well known!!

In addition this is a single institution, retrospective study with all the limitations associated with it!!

Comments:

Point 1.

How do you know all lung cancers behave similarly in terms of NLR, PLR and MLR?? The analysis of different lung types (adenoCa, squamous, NOS) which possess different molecular and behavioral characteristics is methodologically not correct!!

AU: Thank you for your comment. We have clearly stated in limitations of the study that our study presents with inherent weaknesses of a retrospective real-life study, not designed to provide mechanistic data. We also added that: “Nevertheless, real-world studies can provide complementary data on prognostic biomarkers beyond highly-selective randomized controlled trial patient populations.”

In order to be methodologically precise and inclusive, we added in our Cox-regression analysis additional confounders including histopathology, molecular mutations and PD-L1 expression.

Point 2.

I don't share the authors' opinion that bioprofiles such as NLR, PLR, MLR, CRP, WCC are predictive of survival and outcomes!! And at what stage? Pre-op., pre-chemo, after chemo??? Please explain more / defend the concept more although I am uncertain if the evidence are supportive!

AU: Thank you for your comment. In our study, we did not investigate the predictive role of CBC parameters in association with received treatment, as our study was not designed to assess the theragnostic role of peripheral blood biomarkers.   Our aim was to investigate the prognostic role of these parameters at first diagnosis of LC prior to any therapeutic intervention. The fact that we embedded recent relevant references with promising results is supportive of the concept of the study.

Point 3.

Was histopathological diagnosis obtained through bronchoscopy/EBUS/FNB confirmed with surgical histopathology? Often they don't match!!

AU: Thank you for your comment.  With all due respect to your expertise, our data do not agree with your observation. Histopathological diagnosis through EBUS or CT-guided FNB is absolutely reliable with no further need for surgical histopathology confirmation. Based on recent NCCN guidelines, methods for evaluation include mediastinoscopy, mediastinotomy, EBUS, EUS, and CT-guided biopsy. Moreover, multidisciplinary evaluation that includes treating physicians and specialists in obtaining tissue diagnosis (thoracic surgery, interventional pulmonology, and interventional radiology) insists on using the safest and most efficient approach for biopsy.

Point 4.

The 353 patients included in the study seem small for such a complex study! Was a sample size calculation/power analysis performed prior to data mining/patient recruitment?

AU: Thank you for your comment. We have clearly stated in limitations of the study that our sample size is moderate compared to previous reports for the prognostic role of CBC rates; yet, the size is acceptable for a real-life study.

A power analysis was not performed as it was a retrospective real-world study. Sample size calculating is usually conducted in prospective random control studies. 

Point 5.

Because there are a number of confounding variables that can affect survival in all these non-homogenous group of patients the model should also include histopathological type, type of chemo/immunotherapy given, lung resection surgery or not!!

AU: Thank you for your comment. We further adjusted Cox regression model by adding histopathology. With regards to treatment received, we clarified in study design section that all patients were treated based on multidisciplinary discussion between oncologists, pulmonologists, pathologists, radiation oncologists and thoracic surgeons, taking into account the current treatment guidelines at the time of diagnosis.

Point 6.

A table with NLR, PLR, MLR values should be included maybe tabulated by histology or stage!!

AU: Thank you for your comment. We added a table (Table 2) with subgroup NLR, PLR, MLR median values based on histology and stage.

Point 7.

The included images are small and difficult to read.

AU: Thank you for your comment. We corrected it.

Point 8.

How were the NLR, PLR and MLR cut off values determined? This is not adequately explained. Also why use a cut off since these are continuous variables?

AU: Thank you for your comment. In statistical analysis section, we have mentioned that we examined the prognostic accuracy of median levels of studied parameters (NLR, PLR, MLR) in differentiating high from low risk groups of patients with LC in terms of all-cause mortality. More specifically, patients were split based on the median value of each parameter into high and low risk groups. Mortality risk was compared between high and low risk groups. Continuous data can be dichotomized when deciding a cut-off for diagnostic criteria.

Point 9.

How could stage be measured in a linear axis with 2.0, 2.5, 3.0. 3.5 ect!! How are those values associated with stage?  

AU: Thank you for your comment. We apologize for the misunderstanding. We corrected the figure 2.

Reviewer 4 Report

Comments and Suggestions for Authors

Dear Editor and Authors,

Thank you for asking me to review this manuscript titled "The Prognostic Role of Neutrophil to Lymphocyte Ratio, Platelet to Lymphocyte Ratio, and Monocyte to Lymphocyte Ratio in Patients with Lung Cancer at Diagnosis" by Dr. Ourania Papaioannou and colleagues from various institutions in Greece.

The whole concept of NLR, PLR and MLR in relation to cancer outcome has been done at nauseum!! with multiple and larger studies having been reported. The outcomes of this study not surprisingly are quite common and well known!!

In addition this is a single institution, retrospective study with all the limitations associated with it!!

Comments:

  1. How do you know all lung cancers behave similarly in terms of NLR, PLR and MLR?? The analysis of different lung types (adenoCa, squamous, NOS) which possess different molecular and behavioral charachteristics is methodologically not correct!!
  2. I don't share the authors' opinion that bioprofiles such as NLR, PLR, MLR, CRP, WCC are predictive of survival and outcomes!! And at what stage? Pre-op., pre-chemo, after chemo??? Please explain more / defend the concept more although I am uncertain if the evidence are supportive!
  3. Was histopathological diagnosis obtained through bronchoscopy/EBUS/FNB confirmed with surgical histopathology? Often they don't match!!
  4. The 353 patients included in the study seem small for such a complex study! Was a sample size calculation/power analysis performed prior to data mining/patient recruitment?
  5. Because there are a number of confounding variables that can affect survival in all these non-homogenous group of patients the model should also include histopathological type, type of chemo/immunotherapy given, lung resection surgery or not!!
  6. A table with NLR, PLR, MLR values should be included maybe tabulated by histology or stage!!
  7. The included images are small and difficult to read.
  8. How were the NLR, PLR and MLR cut off values determined? This is not adequately explained. Also why use a cut of since these are continous variables?
  9. How could stage be measured in a linear axis with 2.0, 2.5, 3.0. 3.5 ect!! How are those values associated with stage?                           

In conclusion, this study has a wide scope and examines a non-homogeneous group of patients with multiple co-founding factors that can affect outcome which have not been statistically adjusted adequately!! I maintain it is of limitted value at its current form!!

Author Response

Thank you for your valuable comments and the fast peer-review process. Comments really helped us to improve the quality of our manuscript. 

Our point-by-point reply to reviewers follows:

Reviewer: 4

In this manuscript, the authors have performed a retrospective study on the association between lung cancer and increased NLR and PLR. the study was performed on 357 patients. The authors analyzed the blood parameters and the lung cancer stages. The idea is good but the authors need more evidence to support the association study. Considering that the current study is a retrospective study, I would suggest the authors to discuss the implication of this correlation in real life scenario where the lung cancer is not detected yet. In many other diseases also there is an increase in PLR. In this context, I would suggest the authors to discuss the relevance of the study. Although the findings are not novel, the findings will be useful for the society. My comments are provided below

Minor

Point 1.

The authors need to elaborate on the figure legend. The Table 1 is fine but all the figures need elaboration.

AU: Thank you for your comment. We embedded revised figures with more detail.

Major

Point 1.

In figure 1, the difference between two groups (for NLR and PLR) is very small. Although the statistical significance is quite strong, I would suggest the authors to discuss the biological relevance of this small difference.

AU: Thank you for your really nice comment. We agree that the vertical gap between the aforementioned curves decreases as time goes by.  A small vertical gap means a small difference in the proportion of subjects who have survived up to that time, confirmed by the number at risk mentioned below the figures. We clarified it in figure 1, as well.

Point 2.

In Figure 3, the authors need to provide the rationale for studying the association between smoking and NLR.

AU: Thank you for your comment. We added the following  sentence in the discussion section of our manuscript to clarify the rationale: “With regards to the association between smoking and NLR and PLR, the medical literature has shown that NLR increases in correlation with pack-years while PLR is not affected or decreased in smokers.”

Point 3.

The Figure 4 needs elaboration. What is the biological meaning of the highlighted p value in Fig 4?

AU: Thank you for your comment. We clarified in figure 4 that:  “Following adjustment for age, gender, smoking, TNM staging, histopathology, molecular driver mutations and PD-L1 expression, the multivariate model revealed that increased NLR and advanced TNM staging were independent risk factors for mortality.”

Point 4.

The authors have discussed the limitation of the study but the authors did not discuss its relevance in many other diseases. Is this specific for lung cancer. What is the exclusion criteria? Is there any comorbidity?

AU: Thank you for your comment. We apologize for not mentioning this. With regards to exclusion criteria, we clarified in study design section that patients with active infection confirmed through clinical, laboratory or radiological examination at the time of diagnosis were excluded, as well as patients with uncontrolled cardiovascular or pulmonary disease, endocrinological disorders and history of other type of cancer.

We also added the following in introduction section:

“Inflammation has been suggested as prognostic biomarker in cancer due to its contribution as hallmark in tumor microenvironment. Immune  cells are increasingly accepted to be major components of tumor lesions. These inflammatory cells operate in conflicting directions, either antagonizing or promoting neoplastic lesions. As a consequence, systemic inflammation has been implicated in various stages of carcinogenesis, including the initiation, progression, invasion, and metastatic phases, highlighting its potential role as a key prognostic determinant in cancer. Despite the fact that several studies have discussed the prognostic importance of the acute phase reactants in systemic inflammation, including lactate dehydrogenase (LDH), erythrocyte sedimentation rate, alanine transaminase (ALT), aspartate transaminase (AST), interleukins, and C-reactive protein (CRP), none of these parameters are specific to carcinogenesis. On the other side, systemic inflammation can be estimated on a regular basis through simple, inexpensive, easily accessible, and clinicians’ friendly  inflammatory markers based on the complete blood count (CBC) parameters such as neutrophil to lymphocyte ratio (NLR), platelet to lymphocyte ratio (PLR), and monocyte to lymphocyte ratio (MLR). These ratios are recognized as independent prognostic factors in several types of cancer, particularly lung and breast cancer.”

Round 2

Reviewer 2 Report

Comments and Suggestions for Authors

Authors have addressed all the comments, and now revised version looks good, can be accepted for publication 

Reviewer 3 Report

Comments and Suggestions for Authors

The authors have addressed all my concern in the revised manuscript. I support the publication of the revised manuscript.

Reviewer 4 Report

Comments and Suggestions for Authors

Dear Editor and Authors,

I re-reviewed and re-evaluated the revised manuscript the authors have re-submitted. They have edited the paper appropriately and addressed most of this reviewer's comments. Therefore, I am now happy to recommend its publication. Good job to the authors. 

Kind regards to all.